# A General Framework for Robust Interactive Learning*

**Ehsan Emamjomeh-Zadeh**[†]        David Kempe[‡]

## Abstract

We propose a general framework for interactively learning models, such as (binary or non-binary) classifiers, orderings/rankings of items, or clusterings of data points. Our framework is based on a generalization of Angluin's equivalence query model and Littlestone's online learning model: in each iteration, the algorithm proposes a model, and the user either accepts it or reveals a specific mistake in the proposal. The feedback is correct only with probability $p > \frac{1}{2}$ (and adversarially incorrect with probability $1 - p$), i.e., the algorithm must be able to learn in the presence of arbitrary noise. The algorithm's goal is to learn the ground truth model using few iterations.

Our general framework is based on a graph representation of the models and user feedback. To be able to learn efficiently, it is sufficient that there be a graph $G$ whose nodes are the models, and (weighted) edges capture the user feedback, with the property that if $s, s^*$ are the proposed and target models, respectively, then any (correct) user feedback $s'$ must lie on a shortest $s$-$s^*$ path in $G$. Under this one assumption, there is a natural algorithm, reminiscent of the Multiplicative Weights Update algorithm, which will efficiently learn $s^*$ even in the presence of noise in the user's feedback.

From this general result, we rederive with barely any extra effort classic results on learning of classifiers and a recent result on interactive clustering; in addition, we easily obtain new interactive learning algorithms for ordering/ranking.

## 1   Introduction

With the pervasive reliance on machine learning systems across myriad application domains in the real world, these systems frequently need to be deployed before they are fully trained. This is particularly true when the systems are supposed to learn a specific user's (or a small group of users') personal and idiosyncratic preferences. As a result, we are seeing an increased practical interest in online and interactive learning across a variety of domains.

A second feature of the deployment of such systems "in the wild" is that the feedback the system receives is likely to be noisy. Not only may individual users give incorrect feedback, but even if they do not, the preferences — and hence feedback — across different users may vary. Thus, interactive learning algorithms deployed in real-world systems must be resilient to noisy feedback.

Since the seminal work of Angluin [2] and Littlestone [14], the paradigmatic application of (noisy) interactive learning has been online learning of a binary classifier when the algorithm is provided with feedback on samples it had previously classified incorrectly. However, beyond (binary or other) classifiers, there are many other models that must be frequently learned in an interactive manner. Two

[†]Department of Computer Science, University of Southern California, emamjome@usc.edu

[‡]Department of Computer Science, University of Southern California, dkempe@usc.edu

particularly relevant examples are the following: (1) Learning an ordering/ranking of items is a key part of personalized Web search or other information-retrieval systems (e.g., [12, 18]). The user is typically presented with an ordering of items, and from her clicks or lack thereof, an algorithm can infer items that are in the wrong order. (2) Interactively learning a clustering [6, 5, 4] is important in many application domains, such as interactively identifying communities in social networks or partitioning an image into distinct objects. The user will be shown a candidate clustering, and can express that two clusters should be merged, or a cluster should be split into two.

In all three examples — classification, ranking, and clustering — the interactive algorithm proposes a *model*[4] (a classifier, ranking, or clustering) as a solution. The user then provides — explicitly or implicitly — feedback on whether the model is correct or needs to be fixed/improved. This feedback may be incorrect with some probability. Based on the feedback, the algorithm proposes a new and possibly very different model, and the process repeats. This type of interaction is the natural generalization of Angluin's equivalence query model [2, 3]. It is worth noting that in contrast to active learning, in interactive learning (which is the focus of this work), the algorithm cannot "ask" direct questions; it can only propose a model and receive feedback in return. The algorithm should minimize the number of user interactions, i.e., the number of times that the user needs to propose fixes. A secondary goal is to make the algorithm's internal computations efficient as well.

The main contribution of this article is a general framework for efficient interactive learning of models (even with noisy feedback), presented in detail in Section 2. We consider the set of all $N$ models as nodes of a positively weighted undirected or directed graph $G$. The one key property that $G$ must satisfy is the following: (*) If $s$ is a proposed model, and the user (correctly) suggests changing it to $s'$, then the graph must contain the edge $(s, s')$; furthermore, $(s, s')$ must lie on a shortest path from $s$ to the target model $s^*$ (which is unknown to the algorithm).

We show that this single property is enough to learn the target model $s^*$ using at most $\log N$ queries[5] to the user, in the absence of noise. When the feedback is correct with probability $p > \frac{1}{2}$, the required number of queries gracefully deteriorates to $O(\log N)$; the constant depends on $p$. We emphasize that the assumption (*) is not an assumption on the user. We do not assume that the user somehow "knows" the graph $G$ and computes shortest paths in order to find a response. Rather, (*) states that $G$ was correctly chosen to model the underlying domain, so that correct answers by the user must in fact have the property (*). To illustrate the generality of our framework, we apply it to ordering, clustering, and classification:

1. For ordering/ranking, each permutation is a node in $G$; one permutation is the unknown target. If the user can point out only *adjacent* elements that are out of order, then $G$ is an adjacent transposition "BUBBLE SORT" graph, which naturally has the property (*). If the user can pick any element and suggest that it should precede an entire block of elements it currently follows, then we can instead use an "INSERSION SORT" graph; interestingly, to ensure the property (*), this graph must be weighted. On the other hand, as we show in Section 3, if the user can propose two arbitrary elements that should be swapped, there is *no* graph $G$ with the property (*).
   Our framework directly leads to an interactive algorithm that will learn the correct ordering of $n$ items in $O(\log(n!)) = O(n \log n)$ queries; we show that this bound is optimal under the equivalence query model.

2. For learning a clustering of $n$ items, the user can either propose merging two clusters, or splitting one cluster. In the interactive clustering model of [6, 5, 4], the user can specify *that* a particular cluster $C$ should be split, but does not give a specific split. We show in Section 4 that there is a weighted directed graph with the property (*); then, if each cluster is from a "small" concept class of size at most $M$ (such as having low VC-dimension), there is an algorithm finding the true clustering in $O(k \log M)$ queries, where $k$ is number of the clusters (known ahead of time).

3. For binary classification, $G$ is simply an $n$-dimensional hypercube (where $n$ is the number of sample points that are to be classified). As shown in Section 5, one immediately recovers a close variant of standard online learning algorithms within this framework. An extension to classification with more than two classes is very straightforward.

Due to space limits, all proofs and several other details and discussions are omitted. A full version is available on the arXiv at `https://arxiv.org/abs/1710.05422`.

## 2 Learning Framework

We define a framework for query-efficient interactive learning of different types of *models*. Some prototypical examples of models to be learned are rankings/orderings of items, (unlabeled) clusterings of graphs or data points, and (binary or non-binary) classifiers. We denote the set of all candidate models (permutations, partitions, or functions from the hypercube to $\{0, 1\}$) by $\Sigma$, and individual models[6] by $s, s', s^*$, etc. We write $N = |\Sigma|$ for the number of candidate models.

We study interactive learning of such models in a natural generalization of the equivalence query model of Angluin [2, 3]. This model is equivalent to the more widely known online learning model of Littlestone [14], but more naturally fits the description of user interactions we follow here. It has also served as the foundation for the interactive clustering model of Balcan and Blum [6] and Awasthi et al. [5, 4].

In the *interactive learning framework*, there is an unknown ground truth model $s^*$ to be learned. In each round, the learning algorithm proposes a model $s$ to the user. In response, with probability $p > \frac{1}{2}$, the user provides correct feedback. In the remaining case (i.e., with probability $1 - p$), the feedback is *arbitrary*; in particular, it could be arbitrarily and deliberately misleading.

Correct feedback is of the following form: if $s = s^*$, then the algorithm is told this fact in the form of a user response of $s$. Otherwise, the user reveals a model $s' \neq s$ that is "more similar" to $s^*$ than $s$ was. The exact nature of "more similar," as well as the possibly restricted set of suggestions $s'$ that the user can propose, depend on the application domain. Indeed, the strength of our proposed framework is that it provides strong query complexity guarantees under minimal assumptions about the nature of the feedback; to employ the framework, one merely has to verify that the the following assumption holds.

**Definition 2.1 (Graph Model for Feedback)** *Define a weighted graph $G$ (directed or undirected) that contains one node for each model $s \in \Sigma$, and an edge $(s, s')$ with arbitrary positive edge length $\omega_{(s,s')} > 0$ if the user is allowed to propose $s'$ in response to $s$. (Choosing the lengths of edges is an important part of using the framework.) $G$ may contain additional edges not corresponding to any user feedback. The key property that $G$ must satisfy is the following: (\*) If the algorithm proposes $s$ and the ground truth is $s^* \neq s$, then every correct user feedback $s'$ lies on a shortest path from $s$ to $s^*$ in $G$ with respect to the lengths $\omega_e$. If there are multiple candidate nodes $s'$, then there is no guarantee on which one the algorithm will be given by the user.*

### 2.1 Algorithm and Guarantees

Our algorithms are direct reformulations and slight generalizations of algorithms recently proposed by Emamjomeh-Zadeh et al. [10], which itself was a significant generalization of the natural "Halving Algorithm" for learning a classifier (e.g., [14]). They studied the search problem as an abstract problem they termed "Binary Search in Graphs," without discussing any applications. Our main contribution here is the application of the abstract search problem to a large variety of interactive learning problems, and a framework that makes such applications easy. We begin with the simplest case $p = 1$, i.e., when the algorithm only receives correct feedback.

Algorithm 1 gives essentially best-possible general guarantees [10]. To state the algorithm and its guarantees, we need the notion of an approximate median node of the graph $G$. First, we denote by

$$N(s, s') := \begin{cases} \{s\} & \text{if } s' = s \\ \{\hat{s} \mid s' \text{ lies on a shortest path from } s \text{ to } \hat{s}\} & \text{if } s' \neq s \end{cases}$$

the set of all models $\hat{s}$ that are consistent with a user feedback of $s'$ to a model $s$. In anticipation of the noisy case, we allow models to be weighted[7], and denote the node weights or *likelihoods* by

$\mu(s) \geq 0$. If feedback is not noisy (i.e., $p = 1$), all the non-zero node weights are equal. For every subset of models $S$, we write $\mu(S) := \sum_{s \in S} \mu(s)$ for the total node weight of the models in $S$. Now, for every model $s$, define

$$\Phi_\mu(s) := \frac{1}{\mu(\Sigma)} \cdot \max_{s' \neq s, (s,s') \in G} \mu(N(s,s'))$$

to be the largest fraction (with respect to node weights) of models that could still be consistent with a worst-case response $s'$ to a proposed model of $s$. For every subset of models $S$, we denote by $\mu_S$ the likelihood function that assigns weight 1 to every node $s \in S$ and 0 elsewhere. For simplicity of notation, we use $\Phi_S(s)$ when the node weights are $\mu_S$.

The simple key insight of [10] can be summarized and reformulated as the following proposition:

**Proposition 2.1 ([10], Proofs of Theorems 3 and 14)** *Let $G$ be a (weighted) directed graph in which each edge $e$ with length $\omega_e$ is part of a cycle of total edge length at most $c \cdot \omega_e$. Then, for every node weight function $\mu$, there exists a model $s$ such that $\Phi_\mu(s) \leq \frac{c-1}{c}$.*

*When $G$ is undirected (and hence $c = 2$), for every node weight function $\mu$, there exists an $s$ such that $\Phi_\mu(s) \leq \frac{1}{2}$.*

In Algorithm 1, we always have uniform node weight for all the models which are consistent with all the feedback received so far, and node weight 0 for models that are inconsistent with at least one response. Prior knowledge about candidates for $s^*$ can be incorporated by providing the algorithm with the input $S_{\text{init}} \ni s^*$ to focus its search on; in the absence of prior knowledge, the algorithm can be given $S_{\text{init}} = \Sigma$.

---

**Algorithm 1** LEARNING A MODEL WITHOUT FEEDBACK ERRORS $(S_{\text{init}})$

---

1: $S \leftarrow S_{\text{init}}$.
2: **while** $|S| > 1$ **do**
3:     Let $s$ be a model with a "small" value of $\Phi_S(s)$.
4:     Let $s'$ be the user's feedback model.
5:     Set $S \leftarrow S \cap N(s,s')$.
6: **return** the only remaining model in $S$.

---

Line 3 is underspecified as "small." Typically, an algorithm would choose the $s$ with smallest $\Phi_S(s)$. But computational efficiency constraints or other restrictions (see Sections 2.2 and 5) may preclude this choice and force the algorithm to choose a suboptimal $s$. The guarantee of Algorithm 1 is summarized by the following Theorem 2.2. It is a straightforward generalization of Theorems 3 and 14 from [10]

**Theorem 2.2** *Let $N_0 = |S_{init}|$ be the number of initial candidate models. If each model $s$ chosen in Line 3 of Algorithm 1 has $\Phi_S(s) \leq \beta$, then Algorithm 1 finds $s^*$ using at most $\log_{1/\beta} N_0$ queries.*

**Corollary 2.3** *When $G$ is undirected and the optimal $s$ is used in each iteration, $\beta = \frac{1}{2}$ and Algorithm 1 finds $s^*$ using at most $\log_2 N_0$ queries.*

In the presence of noise, the algorithm is more complicated. The algorithm and its analysis are given in the full version. The performance of the robust algorithm is summarized in Theorem 2.4.

**Theorem 2.4** *Let $\beta \in [\frac{1}{2}, 1)$, define $\tau = \beta p + (1 - \beta)(1 - p)$, and let $N_0 = |S_{init}|$. Assume that $\log(1/\tau) > H(p)$ where $H(p) = -p \log p - (1 - p) \log(1 - p)$ denotes the entropy. (When $\beta = \frac{1}{2}$, this holds for every $p > \frac{1}{2}$.)*

*If in each iteration, the algorithm can find a model $s$ with $\Phi_\mu(s) \leq \beta$, then with probability at least $1 - \delta$, the robust algorithm finds $s^*$ using at most $\frac{(1-\delta)}{\log(1/\tau) - H(p)} \log N_0 + o(\log N_0) + O(\log^2(1/\delta))$ queries in expectation.*

**Corollary 2.5** *When the graph $G$ is undirected and the optimal $s$ is used in each iteration, then with probability at least $1 - \delta$, the robust algorithm finds $s^*$ using at most $\frac{(1-\delta)}{1 - H(p)} \log_2 N_0 + o(\log N_0) + O(\log^2(1/\delta))$ queries in expectation.*

## 2.2 Computational Considerations and Sampling

Corollaries 2.3 and 2.5 require the algorithm to find a model $s$ with small $\Phi_\mu(s)$ in each iteration. In most learning applications, the number $N$ of candidate models is exponential in a natural problem parameter $n$, such as the number of sample points (classification), or the number of items to rank or cluster. If computational efficiency is a concern, this precludes explicitly keeping track of the set $S$ or the weights $\mu(s)$. It also rules out determining the model $s$ to query by exhaustive search over all models that have not yet been eliminated.

In some cases, these difficulties can be circumvented by exploiting problem-specific structure. A more general approach relies on Monte Carlo techniques. We show that the ability to sample models $s$ with probability (approximately) proportional to $\mu(s)$ (or approximately uniformly from $S$ in the case of Algorithm 1) is sufficient to essentially achieve the results of Corollaries 2.3 and 2.5 with a computationally efficient algorithm. Notice that both in Algorithm 1 and the robust algorithm with noisy feedback (omitted from this version), the node weights $\mu(s)$ are completely determined by all the query responses the algorithm has seen so far and the probability $p$.

**Theorem 2.6** *Let $n$ be a natural measure of the input size and assume that $\log N$ is polynomial in $n$. Assume that $G = (V, E)$ is undirected[8], all edge lengths are integers, and the maximum degree and diameter (both with respect to the edge lengths) are bounded by $poly(n)$. Also assume w.l.o.g. that $\mu$ is normalized to be a distribution over the nodes[9] (i.e., $\mu(\Sigma) = 1$).*

*Let $0 \leq \Delta < \frac{1}{4}$ be a constant, and assume that there is an oracle that — given a set of query responses — runs in polynomial time in $n$ and returns a model $s$ drawn from a distribution $\mu'$ with $d_{\mathrm{TV}}(\mu, \mu') \leq \Delta$. Also assume that there is a polynomial-time algorithm that, given a model $s$, decides whether or not $s$ is consistent with every given query response or not.*

*Then, for every $\epsilon > 0$, in time $poly(n, \frac{1}{\epsilon})$, an algorithm can find a model $s$ with $\Phi_\mu(s) \leq \frac{1}{2} + 2\Delta + \epsilon$, with high probability.*

# 3 Application I: Learning a Ranking

As a first application, we consider the task of learning the correct order of $n$ elements with supervision in the form of equivalence queries. This task is motivated by learning a user's preference over web search results (e.g., [12, 18]), restaurant or movie orders (e.g., [9]), or many other types of entities. Using pairwise *active* queries ("Do you think that A should be ranked ahead of B?"), a learning algorithm could of course simulate standard $O(n \log n)$ sorting algorithms; this number of queries is necessary and sufficient. However, when using equivalence queries, the user must be presented with a complete ordering (i.e., a permutation $\pi$ of the $n$ elements), and the feedback will be a *mistake* in the proposed permutation. Here, we propose interactive algorithms for learning the correct ranking without additional information or assumptions.[10] We first describe results for a setting with simple feedback in the form of adjacent transpositions; we then show a generalization to more realistic feedback as one is wont to receive in applications such as search engines.

## 3.1 Adjacent Transpositions

We first consider "BUBBLE SORT" feedback of the following form: the user specifies that elements $i$ and $i + 1$ in the proposed permutation $\pi$ are in the wrong relative order. An obvious correction for an algorithm would be to swap the two elements, and leave the rest of $\pi$ intact. This algorithm would exactly implement BUBBLE SORT, and thus require $\Theta(n^2)$ equivalence queries. Our general framework allows us to easily obtain an algorithm with $O(n \log n)$ equivalence queries instead. We define the undirected and unweighted graph $G_{\mathrm{BS}}$ as follows:

- $G_{\mathrm{BS}}$ contains $N = n!$ nodes, one for each permutation $\pi$ of the $n$ elements;
- it contains an edge between $\pi$ and $\pi'$ if and only if $\pi'$ can be obtained from $\pi$ by swapping two adjacent elements.

**Lemma 3.1** $G_{BS}$ *satisfies Definition 2.1 with respect to* BUBBLE SORT *feedback.*

Hence, applying Corollary 2.3 and Theorem 2.4, we immediately obtain the existence of learning algorithms with the following properties:

**Corollary 3.2** *Assume that in response to each equivalence query on a permutation $\pi$, the user responds with an adjacent transposition (or states that the proposed permutation $\pi$ is correct).*

1. *If all query responses are correct, then the target ordering can be learned by an interactive algorithm using at most $\log N = \log n! \leq n \log n$ equivalence queries.*

2. *If query responses are correct with probability $p > \frac{1}{2}$, the target ordering can be learned by an interactive algorithm with probability at least $1 - \delta$ using at most $\frac{(1-\delta)}{1-H(p)} n \log n + o(n \log n) + O(\log^2(1/\delta))$ equivalence queries in expectation.*

Up to constants, the bound of Corollary 3.2 is optimal: Theorem 3.3 shows that $\Omega(n \log n)$ equivalence queries are necessary in the worst case. Notice that Theorem 3.3 does not immediately follow from the classical lower bound for sorting with pairwise comparisons: while the result of a pairwise comparison always reveals one bit, there are $n - 1$ different possible responses to an equivalence query, so up to $O(\log n)$ bits might be revealed. For this reason, the proof of Theorem 3.3 explicitly constructs an adaptive adversary, and does not rely on a simple counting argument.

**Theorem 3.3** *With adversarial responses, any interactive ranking algorithm can be forced to ask $\Omega(n \log n)$ equivalence queries. This is true even if the true ordering is chosen uniformly at random, and only the query responses are adversarial.*

## 3.2 Implicit Feedback from Clicks

In the context of search engines, it has been argued (e.g., by [12, 18, 1]) that a user's clicking behavior provides implicit feedback of a specific form on the ranking. Specifically, since users will typically read the search results from first to last, when a user skips some links that appear earlier in the ranking, and instead clicks on a link that appears later, her action suggests that the later link was more informative or relevant.

Formally, when a user clicks on the element at index $i$, but did not previously click on any elements at indices $j, j+1, \ldots, i-1$, this is interpreted as feedback that element $i$ should precede all of elements $j, j+1, \ldots, i-1$. Thus, the feedback is akin to an "INSERTION SORT" move. (The BUBBLE SORT feedback model is the special case in which $j = i - 1$ always.)

To model this more informative feedback, the new graph $G_{IS}$ has more edges, and the edge lengths are non-uniform. It contains the same $N$ nodes (one for each permutation). For a permutation $\pi$ and indices $1 \leq j < i \leq n$, $\pi_{j \leftarrow i}$ denotes the permutation that is obtained by moving the $i^{\text{th}}$ element in $\pi$ before the $j^{\text{th}}$ element (and thus shifting elements $j, j+1, \ldots, i-1$ one position to the right). In $G_{IS}$, for every permutation $\pi$ and every $1 \leq j < i \leq n$, there is an undirected edge from $\pi$ to $\pi_{j \leftarrow i}$ with length $i - j$. Notice that for $i > j + 1$, there is actually no user feedback corresponding to the edge from $\pi_{j \leftarrow i}$ to $\pi$; however, additional edges are permitted, and Lemma 3.4 establishes that $G_{IS}$ does in fact satisfy the "shortest paths" property.

**Lemma 3.4** $G_{IS}$ *satisfies Definition 2.1 with respect to* INSERTION SORT *feedback.*

As in the case of $G_{BS}$, by applying Corollary 2.3 and Theorem 2.4, we immediately obtain the existence of interactive learning algorithms with the same guarantees as those of Corollary 3.2.

**Corollary 3.5** *Assume that in response to each equivalence query, the user responds with a pair of indices $j < i$ such that element $i$ should precede all elements $j, j+1, \ldots, i-1$.*

1. *If all query responses are correct, then the target ordering can be learned by an interactive algorithm using at most $\log N = \log n! \leq n \log n$ equivalence queries.*

2. *If query responses are correct with probability $p > \frac{1}{2}$, the target ordering can be learned by an interactive algorithm with probability at least $1 - \delta$ using at most $\frac{(1-\delta)}{1-H(p)} n \log n + o(n \log n) + O(\log^2(1/\delta))$ equivalence queries in expectation.*

### 3.3 Computational Considerations

While Corollaries 3.2 and 3.5 imply interactive algorithms using $O(n \log n)$ equivalence queries, they do not guarantee that the internal computations of the algorithms are efficient. The naïve implementation requires keeping track of and comparing likelihoods on all $N = n!$ nodes.

When $p = 1$, i.e., the algorithm only receives correct feedback, it can be made computationally efficient using Theorem 2.6. To apply Theorem 2.6, it suffices to show that one can efficiently sample a (nearly) uniformly random permutation $\pi$ consistent with all feedback received so far. Since the feedback is assumed to be correct, the set of all pairs $(i, j)$ such that the user implied that element $i$ must precede element $j$ must be acyclic, and thus must form a partial order. The sampling problem is thus exactly the problem of sampling a *linear extension* of a given partial order.

This is a well-known problem, and a beautiful result of Bubley and Dyer [8, 7] shows that the Karzanov-Khachiyan Markov Chain [13] mixes rapidly. Huber [11] shows how to modify the Markov Chain sampling technique to obtain an exactly (instead of approximately) uniformly random linear extension of the given partial order. For the purpose of our interactive learning algorithm, the sampling results can be summarized as follows:

**Theorem 3.6 (Huber [11])** *Given a partial order over $n$ elements, let $\mathcal{L}$ be the set of all linear extensions, i.e., the set of all permutations consistent with the partial order. There is an algorithm that runs in expected time $O(n^3 \log n)$ and returns a uniformly random sample from $\mathcal{L}$.*

The maximum node degree in $G_{\text{BS}}$ is $n - 1$, while the maximum node degree in $G_{\text{IS}}$ is $O(n^2)$. The diameter of both $G_{\text{BS}}$ and $G_{\text{IS}}$ is $O(n^2)$. Substituting these bounds and the bound from Theorem 3.6 into Theorem 2.6, we obtain the following corollary:

**Corollary 3.7** *Both under* BUBBLE SORT *feedback and* INSERSION SORT *feedback, if all feedback is correct, there is an* efficient *interactive learning algorithm using at most* $\log n! \leq n \log n$ *equivalence queries to find the target ordering.*

The situation is significantly more challenging when feedback could be incorrect, i.e., when $p < 1$. In this case, the user's feedback is not always consistent and may not form a partial order. In fact, we prove the following hardness result.

**Theorem 3.8** *There exists a $p$ (depending on $n$) for which the following holds. Given a set of user responses, let $\mu(\pi)$ be the likelihood of $\pi$ given the responses, and normalized so that $\sum_{\pi} \mu(\pi) = 1$. Let $0 < \Delta < 1$ be any constant. There is no polynomial-time algorithm to draw a sample from a distribution $\mu'$ with $d_{\text{TV}}(\mu, \mu') \leq 1 - \Delta$ unless $RP = NP$.*

It should be noted that the value of $p$ in the reduction is exponentially close to 1. In this range, incorrect feedback is so unlikely that with high probability, the algorithm will always see a partial order. It might then still be able to sample efficiently. On the other hand, for smaller values of $p$ (e.g., constant $p$), sampling approximately from the likelihood distribution might be possible via a metropolized Karzanov-Khachiyan chain or a different approach. This problem is still open.

## 4 Application II: Learning a Clustering

Many traditional approaches for clustering optimize an (explicit) objective function or rely on assumptions about the data generation process. In interactive clustering, the algorithm repeatedly proposes a clustering, and obtains feedback that two proposed clusters should be merged, or a proposed cluster should be split into two. There are $n$ items, and a *clustering* $\mathcal{C}$ is a partition of the items into disjoint sets (*clusters*) $C_1, C_2, \ldots$. It is known that the target clustering has $k$ clusters, but in order to learn it, the algorithm can query clusterings with more or fewer clusters as well. The user feedback has the following semantics, as proposed by Balcan and Blum [6] and Awasthi et al. [5, 4].

1. MERGE$(C_i, C_j)$: Specifies that all items in $C_i$ and $C_j$ belong to the same cluster.
2. SPLIT$(C_i)$: Specifies that cluster $C_i$ needs to be split, but not into which subclusters.

Notice that feedback that two clusters be merged, or that a cluster be split (when the split is known), can be considered as adding constraints on the clustering (see, e.g., [21]); depending on whether feedback may be incorrect, these constraints are hard or soft.

We define a weighted and *directed* graph $G_{UC}$ on all clusterings $\mathcal{C}$. Thus, $N = B_n \leq n^n$ is the $n^{\text{th}}$ Bell number. When $\mathcal{C}'$ is obtained by a MERGE of two clusters in $\mathcal{C}$, $G_{UC}$ contains a directed edge $(\mathcal{C}, \mathcal{C}')$ of length 2. If $\mathcal{C} = \{C_1, C_2, \ldots\}$ is a clustering, then for each $C_i \in \mathcal{C}$, the graph $G_{UC}$ contains a directed edge of length 1 from $\mathcal{C}$ to $\mathcal{C} \setminus \{C_i\} \cup \{\{v\} \mid v \in C_i\}$. That is, $G_{UC}$ contains an edge from $\mathcal{C}$ to the clustering obtained from breaking $C_i$ into singleton clusters of all its elements. While this may not be the "intended" split of the user, we can still associate this edge with the feedback.

**Lemma 4.1** $G_{UC}$ *satisfies Definition 2.1 with respect to* MERGE *and* SPLIT *feedback.*

$G_{UC}$ is directed, and every edge makes up at least a $\frac{1}{3n}$ fraction of the total length of at least one cycle it participates in. Hence, Proposition 2.1 gives an upper bound of $\frac{3n-1}{3n}$ on the value of $\beta$ in each iteration. A more careful analysis exploiting the specific structure of $G_{UC}$ gives us the following:

**Lemma 4.2** *In* $G_{UC}$, *for every non-negative node weight function* $\mu$, *there exists a clustering* $\mathcal{C}$ *with* $\Phi_\mu(\mathcal{C}) \leq \frac{1}{2}$.

In the absence of noise in the feedback, Lemmas 4.1 and 4.2 and Theorem 2.2 imply an algorithm that finds the true clustering using $\log N = \log B(n) = \Theta(n \log n)$ queries. Notice that this is worse than the "trivial" algorithm, which starts with each node as a singleton cluster and always executes the merge proposed by the user, until it has found the correct clustering; hence, this bound is itself rather trivial.

Non-trivial bounds can be obtained when clusters belong to a restricted set, an approach also followed by Awasthi and Zadeh [5]. If there are at most $M$ candidate clusters, then the number of clusterings is $N_0 \leq M^k$. For example, if there is a set system $\mathcal{F}$ of VC dimension at most $d$ such that each cluster is in the range space of $\mathcal{F}$, then $M = O(n^d)$ by the Sauer-Shelah Lemma [19, 20]. Combining Lemmas 4.1 and 4.2 with Theorems 2.2 and 2.4, we obtain the existence of learning algorithms with the following properties:

**Corollary 4.3** *Assume that in response to each equivalence query, the user responds with* MERGE *or* SPLIT. *Also, assume that there are at most* $M$ *different candidate clusters, and the clustering has (at most)* $k$ *clusters.*

1. *If all query responses are correct, then the target clustering can be learned by an interactive algorithm using at most* $\log N = O(k \log M)$ *equivalence queries. Specifically when* $M = O(n^d)$, *this bound is* $O(kd \log n)$. *This result recovers the main result of [5].*[11]

2. *If query responses are correct with probability* $p > \frac{1}{2}$, *the target clustering can be learned with probability at least* $1 - \delta$ *using at most* $\frac{(1-\delta)k \log M}{1-H(p)} + o(k \log M) + O(\log^2(1/\delta))$ *equivalence queries in expectation. Our framework provides the noise tolerance "for free;" [5] instead obtain results for a different type of noise in the feedback.*

## 5 Application III: Learning a Classifier

Learning a binary classifier is the original and prototypical application of the equivalence query model of Angluin [2], which has seen a large amount of follow-up work since (see, e.g., [16, 17]). Naturally, if no assumptions are made on the classifier, then $n$ queries are necessary in the worst case. In general, applications therefore restrict the concept classes to smaller sets, such as assuming that they have bounded VC dimension. We use $\mathcal{F}$ to denote the set of all possible concepts, and write $M = |\mathcal{F}|$; when $\mathcal{F}$ has VC dimension $d$, the Sauer-Shelah Lemma [19, 20] implies that $M = O(n^d)$.

Learning a binary classifier for $n$ points is an almost trivial application of our framework[12]. When the algorithm proposes a candidate classifier, the feedback it receives is a point with a corrected label (or the fact that the classifier was correct on all points).

We define the graph $G_{\text{CL}}$ to be the $n$-dimensional hypercube[13] with unweighted and undirected edges between every pair of nodes at Hamming distance 1. Because the distance between two classifiers $C$, $C'$ is exactly the number of points on which they disagree, $G_{\text{CL}}$ satisfies Definition 2.1. Hence, we can apply Corollary 2.3 and Theorem 2.4 with $S_{\text{init}}$ equal to the set of all $M$ candidate classifiers, recovering the classic result on learning a classifier in the equivalence query model when feedback is perfect, and extending it to the noisy setting.

**Corollary 5.1**     *1. With perfect feedback, the target classifier is learned using $\log M$ queries[14].*

     *2. When each query response is correct with probability $p > \frac{1}{2}$, there is an algorithm learning the true binary classifier with probability at least $1 - \delta$ using at most $\frac{(1-\delta)\log M}{1 - H(p)} + o(\log M) + O(\log^2(1/\delta))$ queries in expectation.*

## 6   Discussion and Conclusions

We defined a general framework for interactive learning from imperfect responses to equivalence queries, and presented a general algorithm that achieves a small number of queries. We then showed how query-efficient interactive learning algorithms in several domains can be derived with practically no effort as special cases; these include some previously known results (classification and clustering) as well as new results on ranking/ordering.

Our work raises several natural directions for future work. Perhaps most importantly, for which domains can the algorithms be made computationally efficient (in addition to query-efficient)? We provided a positive answer for ordering with perfect query responses, but the question is open for ordering when feedback is imperfect. For classification, when the possible clusters have VC dimension $d$, the time is $O(n^d)$, which is unfortunately still impractical for real-world values of $d$. Maass and Turán [15] show how to obtain better bounds specifically when the sample points form a $d$-dimensional grid; to the best of our knowledge, the question is open when the sample points are arbitrary. The Monte Carlo approach of Theorem 2.6 reduces the question to the question of sampling a uniformly random hyperplane, when the uniformity is over the *partition* induced by the hyperplane (rather than some geometric representation). For clustering, even less appears to be known.

It should be noted that our algorithms may incorporate "improper" learning steps: for instance, when trying to learn a hyperplane classifier, the algorithm in Section 5 may propose intermediate classifiers that are not themselves hyperplanes (though the final output is of course a hyperplane classifier). At an increase of a factor $O(\log d)$ in the number of queries, we can ensure that all steps are proper for hyperplane learning. An interesting question is whether similar bounds can be obtained for other concept classes, and for other problems (such as clustering).

Finally, our noise model is uniform. An alternative would be that the probability of an incorrect response depends on the type of response. In particular, false positives could be extremely likely, for instance, because the user did not try to classify a particular incorrectly labeled data point, or did not see an incorrect ordering of items far down in the ranking. Similarly, some wrong responses may be more likely than others; for example, a user proposing a merge of two clusters (or split of one) might be "roughly" correct, but miss out on a few points (the setting that [5, 4] studied). We believe that several of these extensions should be fairly straightforward to incorporate into the framework, and would mostly lead to additional complexity in notation and in the definition of various parameters. But a complete and principled treatment would be an interesting direction for future work.

**Acknowledgments**

Research supported in part by NSF grant 1619458. We would like to thank Sanjoy Dasgupta, Ilias Diakonikolas, Shaddin Dughmi, Haipeng Luo, Shanghua Teng, and anonymous reviewers for useful feedback and suggestions.

## Footnotes

*A full version is available on the arXiv at `https://arxiv.org/abs/1710.05422`. The present version omits all proofs and several other details and discussions.

[4]We avoid the use of the term "concept," as it typically refers to a binary function, and is thus associated specifically with a classifier.

[5] Unless specified otherwise, all logarithms are base 2.

[6]When considering specific applications, we will switch to notation more in line with that used for the specific application.

[7]Edge lengths are part of the definition of the graph, but node weights will be assigned by our algorithm; they basically correspond to likelihoods.

[8]It is actually sufficient that for every node weight function $\mu : V \to \mathbb{R}^+$, there exists a model $s$ with $\Phi_\mu(s) \leq \frac{1}{2}$.

[9]For Algorithm 1, $\mu$ is uniform over all models consistent with all feedback up to that point.

[10]For example, [12, 18, 9] map items to feature vectors and assume linearity of the target function(s).

[11]In fact, the algorithm in [5] is implicitly computing and querying a node with small $\Phi$ in $G_{UC}$

[12]The results extend readily to learning a classifier with $k \geq 2$ labels.

[13]When there are $k$ labels, $G_{\text{CL}}$ is a graph with $k^n$ nodes.

[14]With $k$ labels, this bound becomes $(k - 1)\log M$.

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
