[Reviews · NeurIPS 2017]

Reviewer 1



The paper proposes a general framework for interactive learning. In the framework, the machine learning models are represented as the nodes in a graph G and the user feedback are represented as weighted edges in G. Under the assumption of “if s, s* are the proposed and target models, then any (correct) user feedback s’ must lie on the shortest s-s* path in G”, the author showed that the Multiplicative Weights Update algorithm can efficiently learn the target model. The framework can be applied to three important machine learning tasks: ranking, clustering, and classification. The problem investigated in the paper is interesting and important. The theoretical results of the paper is convincing. However, these results are not empirically verified on real tasks. My comments on the paper are listed as follows: Pros. 1. A interactive learning framework which models the models and user feedback with graph. The framework can learn the target model in the presence of random noise in the user’s feedback. 2. The framework is really general and can be applied to major machine learning tasks such as classification, clustering and ranking. Cons. 1. My major concern on the paper comes from the absence of empirical evaluation of the proposed framework. The major assumption of the framework is “any (correct) user feedback s’ must lie on the shortest s-s* path in G”. Also, the authors claimed that the proposed framework is robust, which can learn the correct s* even in the presence of random noise in user’s feedback. It is necessary to verify the correctness of these assumption and conclusions based on real data and real tasks. The framework is applied to the tasks of ranking, clustering, and classification, achieving the interactive ranking, interactive clustering, and interactive classification models. It is necessary to compare these derived models with state-of-the-arts models to show the effectiveness of these models. Currently, the authors only showed the theoretical results. 2. There is no conclusion section in the paper. It is better to conclude the contributions of the paper in the last section.

Reviewer 2



The paper introduces a new, elegant framework for interactive learning that generalizes several previous interactive learning frameworks, including the classical equivalence query model of Angluin and the interactive clustering model of Balcan and Blum. Building upon the results of Emamjomeh-Zadeh et al. for their "Binary Search in Graphs" model, this paper gives general meta-algorithms for learning models interactively as well as algorithms for specific problems. The introduced model is elegant and original, and surprisingly flexible to generalize many well-known previous results from disparate fields of learning theory, such as classical results about learning hyperplanes with equivalence queries (by Maass and Turán) or the much more recent results of Awasthi and Zadeh about clustering arbitrary concept classes in the interactive clustering model of Balcan and Blum. The model also yields efficient new algorithms for interactively learning rankings. The paper is excellently written and the introduced model should be of interest to anyone who has worked on various interactive or query frameworks of learning. This new model will likely inspire further research in the area of interactive learning. Although the paper builds heavily on the previous work of Emamjomeh-Zadeh et al. which contains a significant part of the technical machinery used here, the idea of applying Emamjodeh-Zadeh et al.'s results to interactive learning is a novel and clever idea and several of the applications are technically nontrivial. Overall, the paper is a pleasure to read and deserves to be shared with the NIPS community.